# Alternative to Nitric Acid Passivation of 15-5 and 17-4PH Stainless Steel Using Electrochemical Techniques

**DOI:** 10.3390/ma13122836

**Published:** 2020-06-24

**Authors:** María Lara-Banda, Citlalli Gaona-Tiburcio, Patricia Zambrano-Robledo, Marisol Delgado-E, José A. Cabral-Miramontes, Demetrio Nieves-Mendoza, Erick Maldonado-Bandala, Francisco Estupiñan-López, José G. Chacón-Nava, Facundo Almeraya-Calderón

**Affiliations:** 1Universidad Autonoma de Nuevo Leon, FIME—Centro de Investigación e Innovación en Ingeniería Aeronáutica (CIIIA), Av. Universidad s/n. Ciudad Universitaria, San Nicolás de los Garza, Nuevo León 66455, Mexico; marialarabanda@yahoo.com.mx (M.L.-B.); citlalli.gaonatbr@uanl.edu.mx (C.G.-T.); patricia.zambranor@uanl.edu.mx (P.Z.-R.); marisol__1706@hotmail.com (M.D.-E); jose.cabralmr@uanl.edu.mx (J.A.C.-M.); francisco.estupinanlp@uanl.edu.mx (F.E.-L.); 2Facultad de Ingeniería Civil, Universidad Veracruzana, Xalapa, Veracruz 91000, Mexico; dnieves@uv.mx (D.N.-M.); eemalban@gmail.com (E.M.-B.); 3Centro de Investigación en Materiales Avanzados (CIMAV), Miguel de Cervantes 120, Complejo Industrial Chihuahua, Chihuahua, Chih 31136, Mexico; jose.chacon@cimav.edu.mx

**Keywords:** stainless steel, passivated, electrochemical noise, precipitation hardening

## Abstract

Increasingly stringent environmental regulations in different sectors of industry, especially the aeronautical sector, suggest the need for more investigations regarding the effect of environmentally friendly corrosion protective processes. Passivation is a finishing process that makes stainless steels more rust resistant, removing free iron from the steel surface resulting from machining operations. This results in the formation of a protective oxide layer that is less likely to react with the environment and cause corrosion. The most commonly used passivating agent is nitric acid. However, it is know that high levels of toxicity can be generated by using this agent. In this work, a study has been carried out into the electrochemical behavior of 15-5PH (precipitation hardening) and 17-4PH stainless steels passivated with (a) citric and (b) nitric acid solutions for 60 and 90 min at 49 °C, and subsequently exposed to an environment with chlorides. Two electrochemical techniques were used: electrochemical noise (EN) and potentiodynamic polarization curves (PPC) according to ASTM G199-09 and ASTM G5-13, respectively. The results obtained indicated that, for both types of steel, the passive layer formed in citric acid as passivating solution had very similar characteristics to that formed with nitric acid. Furthermore, after exposure to the chloride-containing solution and according with the localization index (LI) values obtained, the stainless steels passivated in citric acid showed a mixed type of corrosion, whereas the steels passivated in nitric acid showed localized corrosion. Overall, the results of the $R_{n}$ values derived show very low and similar corrosion rates for the stainless steels passivated with both citric and nitric acid solutions.

## 1. Introduction

Corrosion in the aeronautical industry remains a major problem that directly affects safety, economic, and logistical issues. Stainless steel alloys have found increasing application in aircraft components that require great strength but can handle the increased weight. The high corrosion and temperature resistances found in stainless steel in harsh environments make it suitable for a range of aircraft parts such as fasteners, actuators and landing gear components [1,2,3]. Passivation is a chemical process to remove surface contamination, i.e., small particles of iron-containing shop dirt and iron particles from cutting tools that can act as initiation sites for corrosion. This process also can remove sulfides exposed on the surface of free-machining stainless alloys. In other words, by chemically removing free contaminants from the surface of stainless steel, the passivation process adds a thin oxide layer. More chromium available from a clean surface means a thicker chromium oxide layer at the top of the stainless steel surface. Moreover, this chemically non-reactive surface means more protection against corrosion [4,5,6,7,8,9,10].

Precipitation hardening (PH) stainless steels (SS) are a family of corrosion resistant alloys some of which can be heat treated to provide tensile strengths of 850 to 1700 MPa and yield strengths of 520 MPa to over 1500 MPa. These alloys contain 11–18% chromium, 3–4% nickel, and smaller counts of additional metals, including aluminum, niobium, molybdenum, titanium and tungsten. Nevertheless, chromium is the alloying element responsible for the formation of the passive film [11,12,13,14]. The family of precipitation hardening stainless steels can be divided into three main types—low carbon martensitic, semi-austenitic and austenitic. These stainless steels are widely used in aerospace structural applications due to its good corrosion resistance and high strength and toughness obtained by the formation of precipitates from age-hardening treatments. Previous investigations on aeronautical-aerospace sector has shown that 15-5PH and 17-4PH steels have good corrosion resistance regarding other stainless steels [15,16,17,18,19,20].

Back in 1997, the specification QQ-P-35 for passivation of stainless steel parts was withdrawn, and replaced by specification SAE-QQ-P-35, also withdrawn in 2005. The latter was replaced by specification ASTM A967-17. This indicates that both, citric and nitric acid can be used as passivating agents for stainless steels. To be effective, the nitric acid must be highly concentrated. However, many questions has been done regarding the production of harmful to health toxic vapors generated by the use of nitric acid in passivation baths [21,22]. On the other hand, citric acid is a biodegradable alternative that does no generate hazardous waste. Although the citric acid benefits as a passivating agent are well-established, technical information about the passivation process is scarce [23,24]. In 2003, Boeing Company evaluate the use of citric acid as an alternative for steel passivation in the aeronautic industry [6]. In 2008, the National Aeronautics and Space Administration (NASA) began a research program focused on the evaluation of the use of nitric acid in the passivation process of welded parts, using the salt chamber technique [10]. Later, NASA evaluated the use of citric acid on specimens exposed under atmospheric corrosion conditions using adherence tests [21].

It is well know that aggressive ions, especially chloride ions Cl-, affect the protecting nature of the passive film on stainless steels causing its breakdown. This leads to localized attack, mainly pitting corrosion [25,26]. In the study of corrosion mechanisms, a number of electrochemical techniques such as potentiodynamic, potentiostatic, and galvanostatic polarization tests, electrochemical impedance (EIS) and electrochemical noise (EN) are widely used. For instance, the evaluation of important parameters such as passive range, pitting potentials, corrosion rates and transpassive regions are studied using potentiodynamic polarization curves (PPC). Bragaglia et al. [27] studied the potentiodynamic polarization behavior of passivated citric and nitric acid baths) and unpassivated AISI 304 stainless steel samples after 1 h in 3.5 wt. % NaCl solution. The passivation treatment largely increased the pitting potential, particularly in the case of nitric acid. After 24 h exposure, electrochemical behavior for the nitric acid and the citric acid passivated samples were almost identical.

Electrochemical noise is a technique that does not alter the natural state of the system, since no external disturbance is applied [28]. This technique reflect random or spontaneous events of current and/or potential fluctuations. Under open-circuit conditions, these fluctuations appear to be related to variations in the rates of anodic and cathodic reactions causing small transients as a result of stochastic processes such as breakdown and repassivation of passive films and formation and propagation of pits. The fluctuations of current between two nominally identical electrodes as well as their potential versus a reference electrode (three electrode system) are recorded as time series, and by using several methods to analyze noise data, an understanding of the corrosion process occurring can be determined. The EN data can be analyzed by several methods. Perhaps the most commonly used are those related to frequency domain (power density spectral or spectral analysis), time domain (statistical methods as skewness, kurtosis, localization index (LI), and the variation of signal amplitude with time) and time-frequency domains [29,30]. Suresh and Mudali [31] studied the corrosion of UNS S30403 stainless steel in 0.05 M ferric chloride (FeCl_3_) by spectral, statistical, and wavelet methods to deduce the corrosion mechanism. They found a good correlation of roll-off slopes derived from power spectral analysis and statistical parameters such as standard deviation, localization index (LI), and kurtosis with pitting as the corrosion mechanism. These authors reported a localization index (LI) in the range from 0.7 and 1. LI values of 0.1 to 1 has been attributed to pitting corrosion and hence the mechanism of corrosion was attributed to pitting attack [X]. Ortiz Alonso et al. [32], studied the stress corrosion cracking (SCC) behavior of a supermartensitic stainless steel by EN. They found that the LI value increased during the straining of specimens (in the range from 0.1 to 1), indicating the presence of localized events such as pits or cracks regardless of the susceptibility of the steel to stress corrosion cracking. In spite of some of its drawbacks, other studies also have found a good relationship between the LI parameter and pitting corrosion [33,34]

The aim of the present work is the study of the electrochemical behavior of 15-5PH and 17-4PH stainless steels passivated in nitric and citric acid and exposed to a 5 wt. % NaCl aqueous solution by PPC and EN.

## 2. Materials and Methods

### 2.1. Materials and Samples Preparation

The materials used in this work were 15-5PH and 17-4PH stainless steels used in the as received condition. The chemical composition of these steels was obtained by atomic absorption spectrometry, see Table 1.

Stainless steel samples were machined as cylindrical coupons, according to ASTM A380-17 [35]. The specimens were polished with SiC grit paper till 4000 grade, followed by ultrasonic cleaning in ethanol and deionized water for about 10 min each.

### 2.2. Passivation Process

The passivation process was carried out under the specification ASTM A967-17 [36]. Gaydos et al. [21] reported that extended passivation treatments give a better protection against corrosion for a series of stainless steels. In the present work, two passivation baths (a) nitric acid (20%v) and (b) citric acid (15%v) solutions were used. A constant temperature of 49 °C was maintained along the passivation process. Specimens were immersed in the solutions for 60 and 90 min. Table 2 show the passivation exposure conditions for each type of steel.

### 2.3. Electrochemical Techniques

In order to assess the corrosion behavior of passivated specimens (exposed area 4.46 cm^2^), two electrochemical techniques were used: EN and PPC. The electrolyte was a 5 wt. % NaCl aqueous solution and all tests were carried out at room temperature.

#### 2.3.1. Electrochemical Noise (EN)

This technique was carried out under ASTM G199-09 standard [37]. The experimental setup for EN measurements is schematically depicted in Figure 1. Here, two nominally identical electrodes (passivated stainless steels) as working electrodes (WE1 and WE2) were connected to measure the electrochemical current noise (ECN), whereas the electrochemical potential noise (EPN) was measured by connecting one working electrode to a saturated calomel reference electrode.

The current and potential electrochemical noise was monitored as a function of time under open circuit condition for each particular electrode–electrolyte combination, using a Gill-AC (Alternating Current) potentiostat/galvanostat/ZRA (Zero Resistance Ammeter) from ACM Instruments. Electrochemical noise measurements started one after the open circuit potential stabilized (about 1 h after immersion in the electrolyte). Since the EN technique involves mostly non-stationary signals, trend removal was carried out. In each experiment, 1024 data were measured with a scanning speed of 1 data/s. The time series in current and potential were visually analyzed to interpret the signal transients and define the behavior of the frequency and amplitude of the fluctuations as a function of time. Resistance noise (*Rn*) data were obtained and used to calculate the corrosion rate according to Equation (1),
(1)Rn=σEσI
where $\sigma_{E}$ is the standar deviation of potential noise, and $\sigma_{I}$ is the standar deviation of current noise after trend removal. The LI, defined by Equation (2), is a parameter used to estimate, as a first approximation, the type of corrosion occurring in a given system [38,39,40]. LI values approaching zero, indicates uniform (general) corrosion; values in the range from 0.01 to 0.1 indicates mixed corrosion, whereas values from 0.1 to 1 correspond to pitting corrosion.
(2)$$IL=\frac{\sigma_{I}}{I_{RMS}}$$
where *I_RMS_* is the root mean square value of the corrosion current noise.

#### 2.3.2. Potentiodynamic Polarization Curves (PPC)

This technique was carried out according to ASTM G5-13 [41] and ASTM G102-89 standards [42]. Here, a conventional three-electrode cell configuration was used, see Figure 2.

Potentiodynamic polarization curves were recorded in 5 wt. % NaCl aqueous solution at room temperature in a Gill-AC potentiostat/galvanostat from ACM Instruments. The potential scan was carried out from −1000 mV to +1200 mV, at a scan rate of 60 mV/min. A saturated calomel electrode (SCE) and a platinum wire were used as reference electrode and counter electrode, respectively. The working electrode (passivated sample) was hold for about 1 h at open circuit potential before tests.

## 3. Results

### 3.1. Electrochemical Noise

Figure 3 and Figure 4 show the current and potential time series recorded for 15-5PH and 17-4PH stainless steel passivated in citric and nitric acid solutions at 60 and 90 min, respectively. Figure 2 shows that under passivation conditions at 60 and 90 min in citric acid, the passivated 15-5PH and 17-4PH stainless steel specimens did not present current fluctuations in time, this indicating that the specimens are in passive conditions; also, the potential noise signals remained constant without fluctuations in time (Figure 3a). The 17-4PH sample passivated for 60 min has higher current demand with low amplitude and high frequency transients, while the potential for this alloy has more active potentials (Figure 3d). For both types of stainless steel, the current-potential time series after 1000 s it has a tendency towards passivation.

Windowing analysis of electrochemical current noise between 0 and 200 s (Figure 3b) show no current increase for the 15-5PH samples passivated at 60 and 90 min. The 17-4PH steel passivated for 60 min, shows some transients of low amplitude and frequency, while for the 90 min passivation treatment only one anodic transient of high amplitude and low frequency was recorded 20 s after the start of the test. Another windowing analysis of current noise signal was performed between 900 and 1024 s (Figure 3c). For both types of stainless steel, irrespective of passivation conditions, no current fluctuations were observed. In some way, this behavior indicates stability of the passive layer.

For both types of stainless steel under passivation conditions, windowing analysis from 0 to 200 s and from 900 to 1024 s did not show frequency or amplitude transients, confirming the stability of potentials (Figure 3e,f). It is worth noting that the potentials of the 17-4PH samples are more negative than those recorded for the 15-5PH samples.

For the 15-5PH and 17-4PH stainless steels passivated in nitric acid, Figure 4 shows the current and potential noise time series recorded. The 17-4PH sample passivated for 90 min, show a decreases in current noise as a function of time; while the potential noise shifts to noble values, indicating stability of the passive layer. A similar behavior was observed for the 15-5PH samples passivated for 60 min. The 15-5PH and 17-4PH samples passivated for 90 and 60 min show a small current demand during the first 300 and 700 s. Afterwards, no significant current or potential fluctuations (transients) were recorded, indicating stabilization of the passive layer (Figure 4a,d).

Windowing analysis of electrochemical current noise between 0 and 200 s (Figure 4b), shows a small increase in in current demand for the 15-5PH and 17-4PH samples passivated for 90 and 60 min, respectively (Figure 4b). From 900 to 1024 s, a windowing analysis of current noise did not show current transients (Figure 4c). Windowing analysis from 0 to 200 s and from 900 to 1024 s did not show frequency or amplitude transients, confirming the stability of potentials (Figure 4e,f). It is interesting to note that, irrespective of the time of passivation treatment, more noble potentials were attained by the 15-5PH stainless steel, in comparison with the 17-4PH steel.

The EN parameters derived from the statistical analysis of current and potential time series measurements are shown in Table 3. The *i_corr_* value obtained from noise resistance (*Rn*) for the samples passivated in citric acid is in the order of 10^−4^ (mA/cm^2^), whereas for the samples passivated in nitric, *i_corr_* values about 10^−5^ (mA/cm^2^) were recorded. The very low values for *i_corr_* obtained for both passivating agents indicate that citric acid could be a potential replacement for nitric acid as passivating agent. Information regarding the type of corrosion that could be occurring is given by the LI parameter. As can be seen from Table 3, the stainless steels passivated in citric acid solution mainly show a mixed corrosion type, whereas the stainless steels passivated in nitric acid solution the LI values indicates localized corrosion.

### 3.2. Potenciodynamic Polarization

The corrosion kinetic behaviour using potentiodynamic polarization can be observed through cathodic and anodic reactions in polarization curves. Corrosion rate in terms of penetration (mm/sec) is one of the main parameters obtained by potentiodynamic polarization curves, according to Faraday’s law (Equation (3)) [40,42,43,44].
(3)$$Corrosion\;rate=K_{1}\frac{i_{corr}}{\delta }\;E.W$$

The potentiodynamic polarization curves obtained for the 15-5PH and 17-4PH stainless steels passivated for 60 min and 90 min in (a) citric acid and (b) nitric acid, and immersed in 5 wt. % NaCl solution are shown in Figure 5.The results for citric acid passivation (Figure 5a) show that the lower E_corr_ value was recorded for the 17-4PH sample passivated for 90 min, while the 15-5PH sample passivated for 90 min has the highest E_corr_. Pitting potentials (E_pitt_) were in the range from 42 mV up to 147 mV. This last value was recorded for the 15-5PH steel passivated for 90 min, this being the best treatment for nitric acid passivation, also corroborated by the lower corrosion rate obtained. For nitric acid passivation conditions, Figure 4b show that the E_pitt_ was largely improved, particularly for the 15-5PH steel passivated for 90 min, and also has the lower corrosion rate in this condition. The lower E_pitt_ value recorded was given by the 17-4PH passivated during 90 min, also giving the highest corrosion rate. On the whole, irrespective of the type of PH stainless steel used, the nitric acid passivation treatment largely increases the pitting potentials compared with the citric acid treatment.

The parameters (E_corr_, E_pitt_, *i_corr_*_,_ and corrosion rate (C.R.)) obtained from the polarization potentiodynamic curves are summarised in Table 4. Very low values of corrosion rate (within the same order of magnitude) were recorded for both 15-5PH and 17-4PH steels, irrespective of the passivation treatment conditions.

## 4. Discussion

Several EN procedures correlating timed dependent fluctuation of current and potential during the corrosion process have been used to indicate the type of corrosion occurring. For instance, it is well recognized that the main source of electrochemical noise is the passive film breakdown process and repassivation process [45,46,47,48,49].

For the passivated 15-5PH and 17-4PH stainless steels in this work, the electrochemical potential time series recorded under nitric acid passivation show a passive region from 0 to 120 mV, whereas for citric acid passivation conditions, the passive region goes from −300 to 25 mV. Thus, passivation in nitric acid occurs at more positive (noble) potentials that in citric acid solutions. To some extent, this might indicate that the passive layer is more stable at more noble potentials [25]. Hence, higher corrosion resistance (Rn) values could be expected for passivation in nitric acid solutions [50]. To some extent, the results of *Rn* in Table 3 confirm this.

As a first approach, to assess the more likely type of corrosion occurring for the 15-5PH and 17-4PH stainless steels passivated in both citric acid and nitric acid solutions, the LI parameter was evaluated from the electrochemical noise data, and was found to be in the range from 0.03 to 0.249, see Table 3. From the LI values obtained for each passivating bath, the corrosion type occurring in citric acid passivation conditions can be attributed to mixed corrosion, whereas for nitric acid passivation conditions, the corrosion type could be attributed to pitting corrosion. LI has been used by several research groups for determining corrosion types under several conditions [51,52,53,54,55,56].

The use of LI to determine corrosion types has been the subject of many discussions among investigators on the data treatment and interpretations using LI [29,30,38,57,58,59]. Since the mean of the noise data (detrended) would be negligible, the standard deviation and root mean square current noise would converge to the same value and, hence, the LI evaluated from the data would be unity, irrespective of the corrosion type. Cottis [60] indicated that LI for identification of localization of corrosion is unduly influenced by the mean current and hence less reliable. In the present study, and as a first attempt, the LI parameter was estimated. Of course, it is recognized that in the study of stainless steels such as those in the present work, investigation of procedures based on the frequency domain and time-frequency domain deserves further attention.

The potentiodynamic polarization curves show passivation behavior for the 15-5PH and 17-4PH steels passivated in citric and nitric acid solutions. The passive zones on stainless steels are commonly made up of primary and secondary zones, which are formed before and after transpassivation, respectively. Potentials above E_pitt_ causes a rapid dissolution [61,62,63]. The passive zone involves the formation of iron and chromium oxide films [61,64,65]. Hence, selective dissolution on the surface of the alloy generates a surface enrichment of Cr^3+^ giving rise to Cr(OH)_3_, as shown in Equation (4). Further dissolution of the hydroxide leads to the formation of a continuum layer of Cr_2_O_3_, according Equation (5) [66,67].
(4)$$Cr^{3+}+3OH^{-}\to \;Cr{\left(OH\right)}_{3\;}+3e^{-}$$
(5)$$Cr{\left(OH\right)}_{3}+Cr+3OH^{-}\;\to \;Cr_{2}O_{3}+3H_{2}O+3e^{-}$$

It has been argued that the anodic reactions during the film growth period are mainly from the oxidation of iron and chromium The following equations indicate the oxidation reactions of iron [68]:(6)$$3Fe+8OH^{-}\;\to \;Fe_{3}O_{4}+4H_{2}O+8e^{-}$$
(7)$$2Fe_{3}O_{4}+2OH^{-}+2H_{2}O\;\to \;6FeOOH+2e^{-}$$
(8)$$2Fe_{3}O_{4}+2OH^{-}\;\to \;3Fe_{2}O_{3}+H_{2}O+2e^{-}$$

For the nitric acid passivation treatment, the transpassive region is above 200 mV vs. ECS, whereas for citric acid passivation conditions, the transpassive region is above 50 mV vs. ECS. The passive film formed under nitric acid passivation conditions has higher E_pitt_ values, in comparison with the E_pitt_ values obtained under citric acid passivation conditions. This fact can be seen as a potential disadvantage for the citric acid treatment. Nevertheless, the corrosion rates obtained for both passivation treatments (Table 4) are very low and similar. Thus, for the PH stainless steel used in this work, citric acid passivation treatments can be as effective as nitric acid passivation treatments.

## 5. Conclusions

In this work, samples of 15-5PH and 17-4PH stainless steel were passivated in (a) citric acid and (b) nitric acid baths and exposed in a 5 wt. % NaCl solution. Their electrochemical behavior was studied by electrochemical noise and potentiodynamic polarization.

EN results show that, for citric solution passivation baths, the stabilization of the passive layer occurs at more active potentials compared to the stabilization potentials for nitric acid passivation baths. From noise resistance (*Rn*) data, very low corrosion rate values were derived for the PH stainless steels passivated in both (citric and nitric) passivating treatments.

Statistical evaluation of the time record was carried out and the localization index (LI) parameter was evaluated. According to the LI results, the PH stainless steels passivated in citric acid solution mainly show a mixed corrosion type, whereas LI values for the PH stainless steels passivated in nitric acid solution indicates localized corrosion.

In general, potentiodynamic polarization results indicated that, irrespective of the type of PH stainless steel used, the nitric acid passivation treatment largely increases the pitting potentials in comparison with the citric acid treatment. Also, for both passivation treatments, very low corrosion rate values (in the order of 10^−7^ mm/year) were recorded for both 15-5PH and 17-4PH steels.

On the whole, citric passivation treatments on PH stainless steels could be a green alternative route to the currently employed nitric passivation treatments.

## Figures and Tables

**Figure 1 materials-13-02836-f001:**
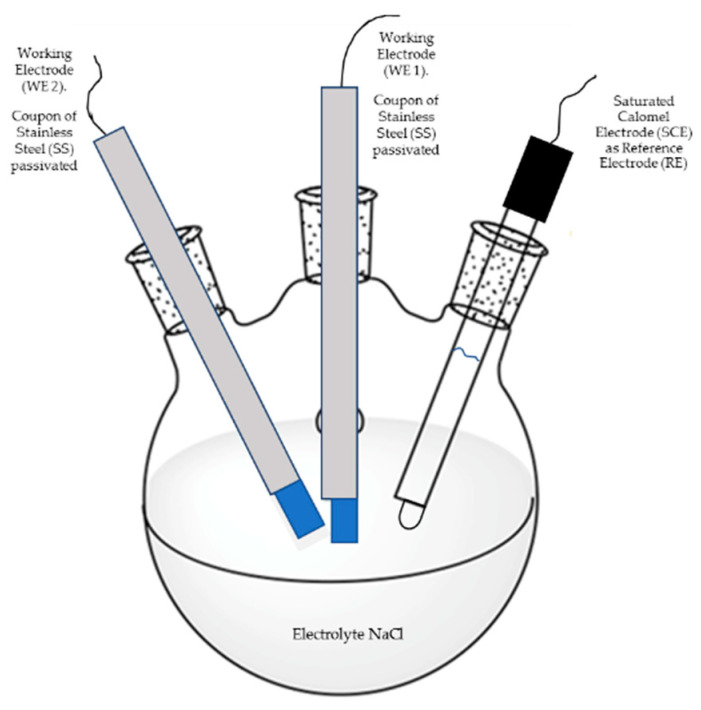
Experimental set up for electrochemical noise (EN) measurements.

**Figure 2 materials-13-02836-f002:**
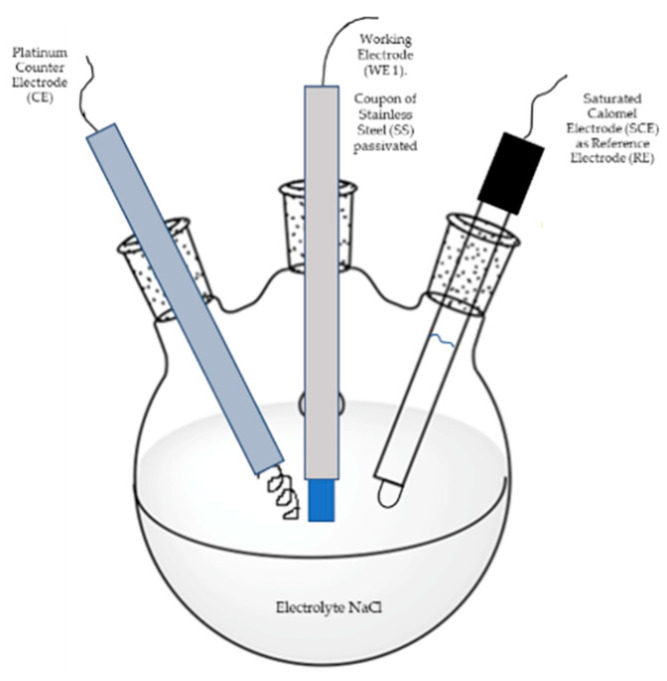
Conventional three-electrode cell configuration used in the potentiodynamic polarization curves (PPC) tests.

**Figure 3 materials-13-02836-f003:**
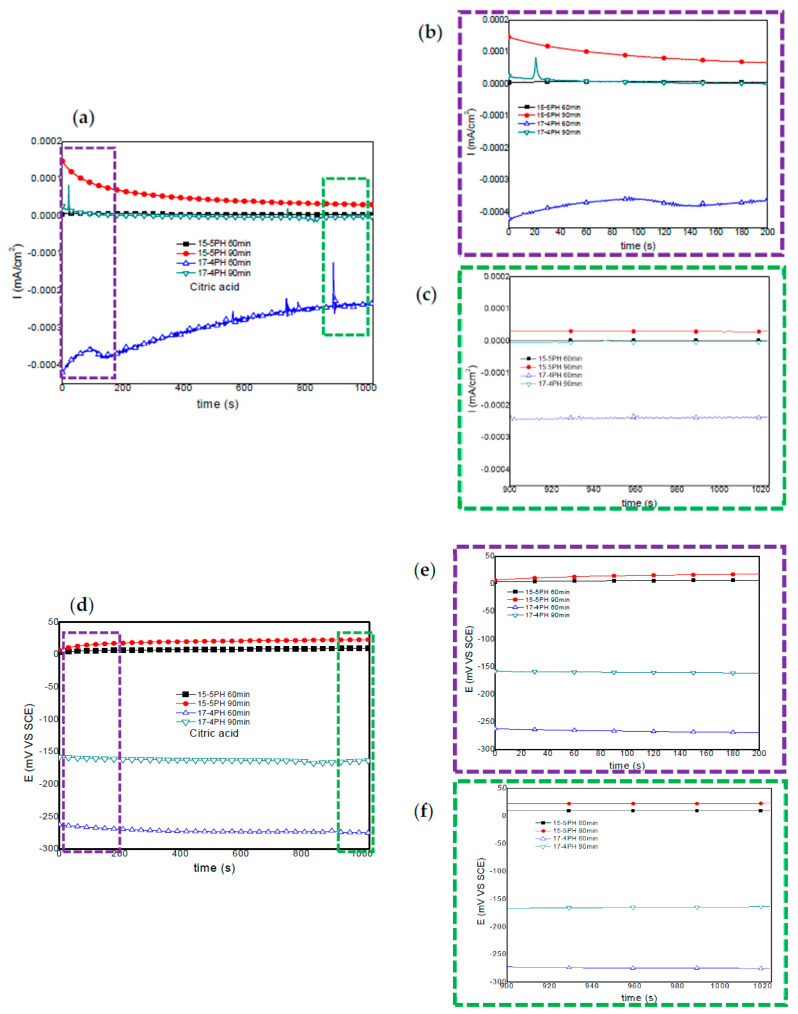
Electrochemical current and potential noise-time series for 15-5PH and 17-4PH samples passivated in citric acid at 49 °C, exposed in a 5 wt.% NaCl solution (**a**,**d**). Windowing of electrochemical current noise (ECN) from 0–200 and 900–1024 s (**b**,**c**); windowing of electrochemical potential noise (EPN) from 0–200 and 900–1024 s (**e**,**f**).

**Figure 4 materials-13-02836-f004:**
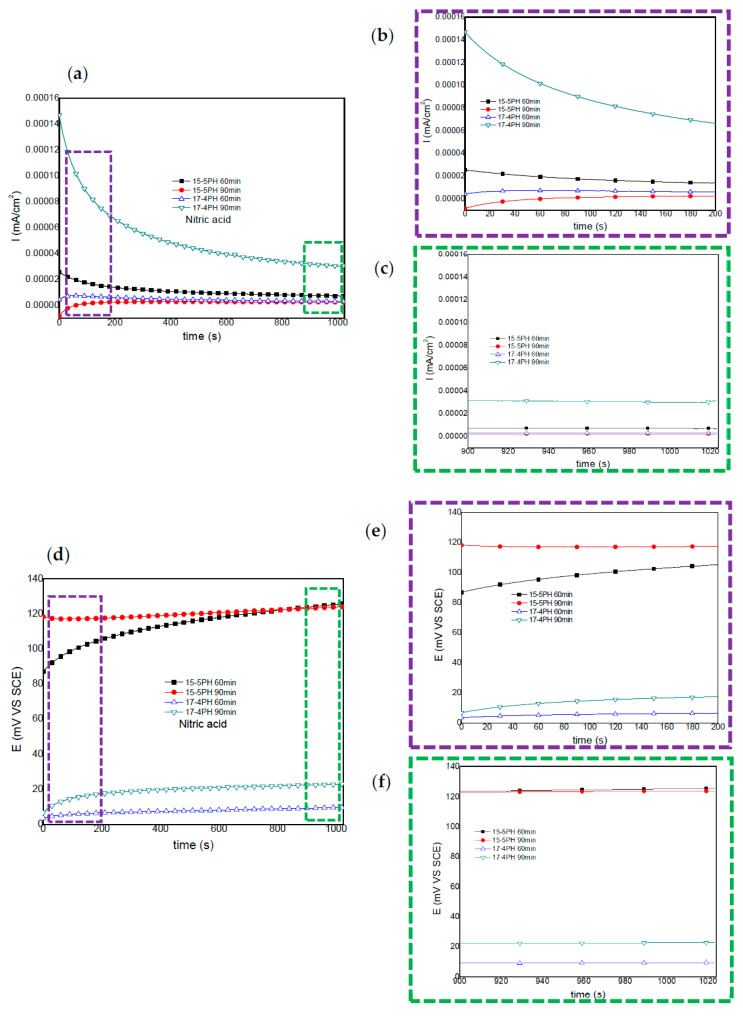
Electrochemical current and potential noise-time series for 15-5PH and 17-4PH samples passivated in nitric acid at 49 °C, exposed in a 5 wt. % NaCl solution (**a**,**d**). Windowing of ECN from 0–200 and 900–1024 s (**b**,**c**); windowing of EPN from 0–200 and 900–1024 s (**e**,**f**).

**Figure 5 materials-13-02836-f005:**
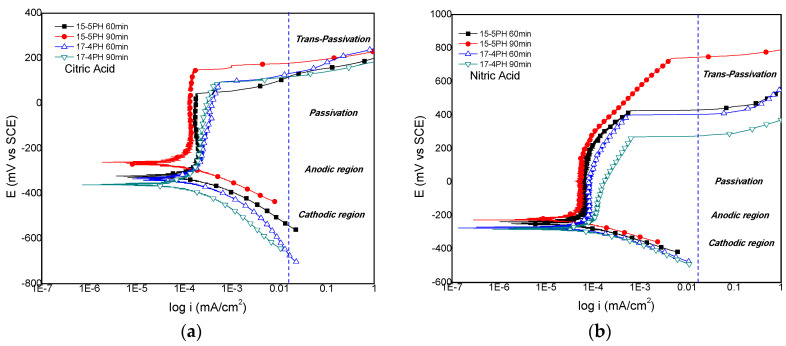
Potentiodynamic polarization curves of 15-5PH and 17-4PH stainless steel passivated in (**a**) citric acid and (**b**) nitric acid, exposed in a 5 wt. % NaCl solution at 49 °C.

**Table 1 materials-13-02836-t001:** Chemical composition of the used stainless steels (wt. %).

Stainless Steel	Elements										
	C	Mn	P	S	Si	Cr	Ni	Mo	Nb	Cu	Fe
15-5PH	0.024	0.817	0.007	0.004	1.569	14.410	3.937	0.383	0.308	3.558	Bal.
17-4PH	0.022	0.827	0.023	0.029	1.637	15.204	3.050	0.340	0.144	3.908	Bal.

**Table 2 materials-13-02836-t002:** Passivation at a temperature of 49 °C.

Stainless Steel	Citric Acid (C_6_H_8_O_7_)		Nitric Acid (HNO_3_)	
	Passivated Time (min)			
	60	90	60	90
15-5PH	X	X	X	X
17-4PH	X	X	X	X

**Table 3 materials-13-02836-t003:** Electrochemical noise parameters at various conditions in 5 wt. % NaCl at 49 °C.

Passivated Agent	Stainless Steel Inoxidable	Time (min)	*Rn* (Ω/cm^2^)	*i_corr_* (mA/cm^2^)	LI	Corrosion Type
Citric acid	15-5PH	60	8.01 × 10^4^	6.49 × 10^4^	0.0862	Mixed
		90	5.00 × 10^5^	1.04 × 10^4^	0.0308	Mixed
	17-4PH	60	5.76 × 10^4^	4.51 × 10^4^	0.2492	Localized
		90	3.27 × 10^5^	1.59 × 10^4^	0.0900	Mixed
Nitric acid	15-5PH	60	2.35 × 10^6^	1.1 × 10^5^	0.1871	Localized
		90	1.51 × 10^6^	1.72 × 10^5^	0.1077	Localized
	17-4PH	60	1.03 × 10^6^	2.52 × 10^5^	0.1485	Localized
		90	1.34 × 10^6^	1.94 × 10^4^	0.1727	Localized

**Table 4 materials-13-02836-t004:** Potentiodynamic polarization parameters in stainless steels passivated at 49 °C, in 5 wt. % NaCl.

Passivated Agent	Stainless Steel	Time (Min)	E_corr_ (mV)	E_pit_ (mV)	*i_corr_* (mA/cm^2^)	C. R. (mm/Year)
Citric Acid	15-5PH	60	−323	42	5.26 × 10^5^	5.54 × 10^7^
		90	−266	147	4.50 × 10^5^	4.75 × 10^7^
	17-4PH	60	−335	91	9.22 × 10^5^	9.64 × 10^7^
		90	−360	97	5.38 × 10^5^	5.63 × 10^7^
Nitric Acid	15-5PH	60	−228	467	2.16 × 10^5^	2.28 × 10^7^
		90	−228	765	2.27 × 10^5^	2.39 × 10^7^
	17-4PH	60	−271	439	3.51 × 10^5^	3.67 × 10^7^
		90	−279	323	4.41 × 10^5^	4.61 × 10^7^

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
