# Peer review of "Alternative to Nitric Acid Passivation of 15-5 and 17-4PH Stainless Steel Using Electrochemical Techniques"

_materials, 2020, doi:10.3390/ma13122836_

Round 1

Reviewer 1 Report

The authors present an interesting work on alternatives to nitric acid passivation of stainless steels which will help in achieving a more environmentally friendly corrosion protection coating. However, the following points below need to be addressed before the manuscript can be accepted.

The manuscript needs to be revised to ensure the grammatical and typographical errors are corrected. Significant portions of the initial write up is not grammatically sound and needs to be revised to ensure the manuscript's aims and procedures are appropriately laid out.

Firstly the title indicates alternative to citric acid while the abstract and manuscript discuss citric acid as an alternative to nitric acid. This needs to be addressed appropriately.

1) The introduction in particular between lines 52 and 58 needs to be rewritten for clarity as it has several grammatical errors that make it difficult to follow the write up.
2) In section 2.3.1 the authors discuss 1024 analysis data. It is unclear what the 1024 refers to analysis data points or empirical measurements here and needs to be edited for clarity.
3) The discussion also needs to be improved to ensure the inferences made are conveyed in a more clear fashion. For example, lines 154 - 157 - while the figures are self-explanatory the discussion is a bit incoherent and needs clarity. Especially the references to Figure 2a and 2d and the corresponding comparisons leave a lot to be desired. In particular it is unclear due to the grammatical structure of the sentences what the discussion is directly referring to. There does not seem to be any major inconsistency from an empirical standpoint but the explanation needs to be fixed to ensure the summary and description are appropriately capturing the trends noted in the experimental results.
4) Figures 2 and 3 have legends that have much smaller fonts than the rest of the manuscript. These need to be fixed so as to allow the readers to discern the data being presented.
5) The write up in lines 276-279 needs to be revised to better describe the trends noted in Figure 4.
6) Reference index needs to be updated in line 313 for Klapper et al.
7) Lines 368-369 need to be edited they refer to nitric acid after breaking down the passive film and refer to why it is better than nitric acid. This needs to be cleaned up to include citric acid and edit appropriately.
Overall, while the technical approach is sound and the empirical dataset has merit for making the presented conclusions, the inherent inconsistencies in the presenetation make it difficult to accept the publication in the current form. This reviewer suggests the authors make appropriate edits to the manuscript to ensure the presentation and the discussion are more coherent and acceptable before a decision can be made on accepting this manuscript.

Author Response

Manuscript ID: Materials-809257

MDPI

 Alternative to Nitric Acid Passivation of 15-5 and 17-4PH Stainless Steel using Electrochemical Techniques

María Lara-Banda1, Citlalli Gaona-Tiburcio1, Patricia Zambrano-Robledo1, Marisol Delgado-E1, José Cabral-Miramontes1, Demetrio Nieves-Mendoza2. Erick Maldonado-Bandala2., Francisco Estupiñan- López1., José G. Chacón-Nava3.,  Facundo Almeraya-Calderón1*

 Reviewer 3

All indications throughout the manuscript were reviewed

Referee Commentaries

Correction attended

The manuscript needs to be revised to ensure the grammatical and typographical errors are corrected. Significant portions of the initial write up is not grammatically sound and needs to be revised to ensure the manuscript's aims and procedures are appropriately laid out.

Attended:

All suggested changes were made in the paper.

Thank you for your comments, the grammar and spelling of the document has been revised in detail.

Firstly the title indicates alternative to citric acid while the abstract and manuscript discuss citric acid as an alternative to nitric acid. This needs to be addressed appropriately.

Attended:

All suggested changes were made in the paper. The title was changed

Alternative to Nitric Acid Passivation of 15-5 and 17-4PH Stainless Steel using Electrochemical Techniques

1) The introduction in particular between lines 52 and 58 needs to be rewritten for clarity as it has several grammatical errors that make it difficult to follow the write up.

Attended:

All suggested changes were made in the paper.

The introduction was redrafted.

Lines 52-66, was improved the wording

2) In section 2.3.1 the authors discuss 1024 analysis data. It is unclear what the 1024 refers to analysis data points or empirical measurements here and needs to be edited for clarity.

Attended:

All suggested changes were made in the paper.

Lines 158-159, was improved the wording. “In each experiment, 1024 data were measured with a scanning speed of 1 data/s”.

3) The discussion also needs to be improved to ensure the inferences made are conveyed in a more clear fashion. For example, lines 154 - 157 - while the figures are self-explanatory the discussion is a bit incoherent and needs clarity. Especially the references to Figure 2a and 2d and the corresponding comparisons leave a lot to be desired. In particular it is unclear due to the grammatical structure of the sentences what the discussion is directly referring to. There does not seem to be any major inconsistency from an empirical standpoint but the explanation needs to be fixed to ensure the summary and description are appropriately capturing the trends noted in the experimental results.

Attended:

All suggested changes were made in the paper.

The grammar and spelling of the results and discussion has been revised in detail.

4) Figures 2 and 3 have legends that have much smaller fonts than the rest of the manuscript. These need to be fixed so as to allow the readers to discern the data being presented.

Attended:

The size of the legends in the figures (3 and 4) were changed.

5) The write up in lines 276-279 needs to be revised to better describe the trends noted in Figure 4.

Attended:

Lines 316-327. The paragraph was rewritten.

The potentiodynamic polarizarion curves obtained for the 15-5PH and 17-4PH stainless steels passivated for 60 min and 90 min in a) citric acid and b) nitric acid, and immersed in 5 %wt NaCl solution are shown in Figure 5.The results for citric acid passivation (Figure 5a) show that the lower Ecorr value was recorded for the 17-4PH sample passivated for 90 min, while the 15-5PH sample passivated for 90 min has the highest Ecorr. Pitting potentials (Epitt) were in the range from 42 mV up to 147 mV. This last value was recorded for the 15-5PH steel passivated for 90 min, being this the best treatment for acid ntric passivation, also corroborated by the lower corrosion rate obtained. For nitric acid passivation conditions, Figure 4b show that the Epitt was largely improved, particularly for the 15-5PH steel passivated for 90 min, and also has the lower corrosion rate in this condition. The lower Epitt value recorded was given by the 17-4PH passivated during 90 min, also giving the highest corrosion rate. On the whole, irrespective of the type of PH stainless steel used, the nitric acid passivation treatment largely increases the pitting potentials compared with the citric acid tratment.

6) Reference index needs to be updated in line 313 for Klapper et al.

Attended:

This is no longer necessary. The paragraph and its reference have been removed.

Some references have been removed and some of the most current have been integrated.

7) Lines 368-369 need to be edited they refer to nitric acid after breaking down the passive film and refer to why it is better than nitric acid. This needs to be cleaned up to include citric acid and edit appropriately.

Attended:

Lines 405-421. All the Conclusions were changed, their spelling and grammar were checked.

·         In this work, samples of 15-5PH and 17-4PH stainless steel were passivated in a) citric acid and b) nitric acid baths and exposed in a 5 %wt NaCl solution. Their electrochemical behavior was studied by electrochemical noise and potentiodynamic polarization.

·         EN results show that, for citric solution passivation baths, the stabilization of the passive layer occurs at more active potentials compared to the stabilization potentials for nitric acid passivation baths. From noise resistance (Rn) data, very low corrosion rate values were derived for the PH stainless steels passivated in both (citric and nitric) passivating treatments.

·         Statistical evaluation of the time record was carried out and the localization index (LI) parameter was evaluated. According to the LI results, the PH stainless steels passivated in citric acid solution mainly show a mixed corrosion type, whereas LI values for the PH stainless steels passivated in nitric acid solution indicates localized corrosion.

·         In general, potentiodynamic polarization results indicated that, irrespective of the type of PH stainless steel used, the nitric acid passivation treatment largely increases the pitting potentials in comparisson with the citric acid treatment. Also, for both passivation treatments, very low corrosion rate values (in the order of 10-7 mm/year) were recorded for both 15-5PH and 17-4PH steels.

·         On the whole, citric passivation treatments on PH stainless steels could be a green alternative route to the currently employed nitric passivation treatments.

Reviewer 2 Report

1.At line 157, authors state that "17-4PH stainless steel passivated 60
minutes has higher current demand with low amplitude and high frequency transients, while the potential of this alloy has more active potentials," The chemical composition have difference between the 17-4PH and The 15-PH steel. Authors should give some further discussion. 

2.At line 182, authors state that "After 100 seconds a stable passive layer is observed, this in contrast to the time series in potential that indicates stability of the passive layer (Figure 3c)."  Where is " a stabe passive layer" in figure 3c ?

3. At line 275, "where can be observed a mixed control bye the activation of anodic and cathodic branch in each case." It may be a misspell. Authors should check again.

4. At line 368, authors state that "Nitric acid after breaking the passive film has a dissolution zone and then the activation of the material occurs, contrary to nitric acid, which after breaking the passive layer is activated." Authors should give some results about the dissolution zone.

Author Response

Manuscript ID: Materials-809257

MDPI

 Alternative to Nitric Acid Passivation of 15-5 and 17-4PH Stainless Steel using Electrochemical Techniques

María Lara-Banda1, Citlalli Gaona-Tiburcio1, Patricia Zambrano-Robledo1, Marisol Delgado-E1, José Cabral-Miramontes1, Demetrio Nieves-Mendoza2. Erick Maldonado-Bandala2., Francisco Estupiñan- López1., José G. Chacón-Nava3.,  Facundo Almeraya-Calderón1*

 Reviewer 1

Referee Commentaries

Correction attended

All indications throughout the manuscript were reviewed and Are marked using green colour letters.

In blue is marked everything that was added as new in the article

References have been removed and some of the most current ones have been integrated

1.      At line 157, authors state that "17-4PH stainless steel passivated 60
minutes has higher current demand with low amplitude and high frequency transients, while the potential of this alloy has more active potentials," The chemical composition have difference between the 17-4PH and The 15-PH steel. Authors should give some further discussion. 

Attended:

The paragraph was corrected, modifying the wording. All suggested changes were made in the paper.

The 17-4PH sample passivated for 60 minutes has higher current demand with low amplitude and high frequency transients, while the potential for this alloy has more active potentials (Figure 3d). For both types of stainless steel, the current-potential time series after 1000 seconds it has a tendency towards passivation. The visual analysis, stainless steels passivated indicate that the shorter the passivation time, they tend to be more active, at the beginning of the test. This may be due to the lower corrosion resistance of 17-4 PH steel, due to its alloying elements that form intermetallics and have a lower concentration than 15-5PH stainless steel.

2.      At line 182, authors state that "After 100 seconds a stable passive layer is observed, this in contrast to the time series in potential that indicates stability of the passive layer (Figure 3c)."  Where is " a stabe passive layer" in figure 3c ?

Attended:

The paragraph was corrected, modifying the wording. All suggested changes were made in the paper.

The 15-5PH and 17-4PH samples passivated for 90 and 60 minutes, show a small current demand during the first 300 and 700 seconds. Afterwards, no significant current or potential fluctuations (transients) were recorded, indicating stabilization of the passive layer. (Figures 4a and 4d).

3.      At line 275, "where can be observed a mixed control bye the activation of anodic and cathodic branch in each case." It may be a misspell. Authors should check again.

Attended:

All suggested changes were made in the paper.

correct "by" instead of "bye". The paragraph has been better drafted.

4.      At line 368, authors state that "Nitric acid after breaking the passive film has a dissolution zone and then the activation of the material occurs, contrary to nitric acid, which after breaking the passive layer is activated." Authors should give some results about the dissolution zone.

Attended:

All suggested changes were made in the paper.

Nitric acid after breaking the passive film has a dissolution zone and then the activation of the material occurs, contrary to nitric acid citric acid, which after breaking the passive layer is activated. there was an error in this paragraph.

Conclusions have been redrafted

EN results show that, for citric solution passivation baths, the stabilization of the passive layer occurs at more active potentials compared to the stabilization potentials for nitric acid passivation baths. From noise resistance (Rn) data, very low corrosion rate values were derived for the PH stainless steels passivated in both (citric and nitric) passivating treatments.

Statistical evaluation of the time record was carried out and the localization index (LI) parameter was evaluated. According to the LI results, the PH stainless steels passivated in citric acid solution mainly show a mixed corrosion type, whereas LI values for the PH stainless steels passivated in nitric acid solution indicates localized corrosion.

In general, potentiodynamic polarization results indicated that, irrespective of the type of PH stainless steel used, the nitric acid passivation treatment largely increases the pitting potentials in comparisson with the citric acid treatment. Also, for both passivation treatments, very low corrosion rate values (in the order of 10-7 mm/year) were recorded for both 15-5PH and 17-4PH steels.

On the whole, citric passivation treatments on PH stainless steels could be a green alternative route to the currently employed nitric passivation treatments.

Reviewer 3 Report

Dear Authors,

I read your article carefully, and unfortunately, I have a lot of serious comments to its preparation.

The main disadvantage of the article is the language which in many sentences or chapters cause a big trouble to understand what was Your idea and message. Besides simple tenses errors some sentences don't have any verb or the sentence is so long and it not provide understandable content.

Check how You named each acid in Table 1.

Chapters are mixed - in Results there are informations which should be in 2 chapter. Additionally I think (but because of the language I'm not sure) that in discussion chapter there are sentences not connected to Your results and better fit to first chapter.

IL is often described as unreliable factor to distinguish corrosion mechanisms. Can You discuss wider why You believe it is enough in this case?

Reference list is very expanded but mostly by quite old articles. There are only 9/54 which are from last 5 years.

Author Response

Manuscript ID: Materials-809257

MDPI

Alternative to Nitric Acid Passivation of 15-5 and 17-4PH Stainless Steel using Electrochemical Techniques

María Lara-Banda1, Citlalli Gaona-Tiburcio1, Patricia Zambrano-Robledo1, Marisol Delgado-E1, José Cabral-Miramontes1, Demetrio Nieves-Mendoza2. Erick Maldonado-Bandala2., Francisco Estupiñan- López1., José G. Chacón-Nava3.,  Facundo Almeraya-Calderón1*

 Reviewer 2

All indications throughout the manuscript were reviewed and Are marked using green colour letters.

In blue is marked everything that was added as new in the article

Referee Commentaries

Correction attended

The main disadvantage of the article is the language which in many sentences or chapters cause a big trouble to understand what was Your idea and message. Besides simple tenses errors some sentences don't have any verb or the sentence is so long and it not provide understandable content.

Attended:

All suggested changes were made in the paper.

Thanks for your comments, the spelling of the paper has been revised in detail.

Check how You named each acid in Table 1.

Attended:

All suggested changes were made in the paper.

The formulas of the acids used have already been corrected in Table 1

Chapters are mixed - in Results there are informations which should be in 2 chapter. Additionally I think (but because of the language I'm not sure) that in discussion chapter there are sentences not connected to Your results and better fit to first chapter.

Attended:

All suggested changes were made in the paper.

The sections of: Abstract, Methodology, results and conclusions have been revised.

Conclusions have been redrafted

The spelling of the paper has been revised in detail.

IL is often described as unreliable factor to distinguish corrosion mechanisms. Can You discuss wider why You believe it is enough in this case?

Attended:

All suggested changes were made in the paper.

      Commentary

For uniform corrosion, the methods of noise analysis using noise resistance and impedance are quite well established; however, the understanding is limited for localized corrosion. Electrochemical noise parameters have been deduced by various investigators to understand localized corrosion [1,2]. Some of the methods that are proposed in literature utilize power spectrum, statistical parameters such as skewness, kurtosis, localization index, bispectrum, or estimation of the intensity of characteristic transient occurrence in voltage, or current records for monitoring pitting corrosion.

After several studies in 1999, Mansfeld and Sun [3] concluded that the localization index can have limitations, so it should be used with caution. Situation that even Eden affirms since in the year of 2001 together with Reid [4] they developed another patent, where the corrosion mechanism is identified from the calculations of kurtosis and skewness, based on statistical moments.

1.      Girija Suresh and U. Kamachi Mudali‡, Electrochemical Noise Analysis of Pitting Corrosion of Type 304L Stainless Steel CORROSION—Vol. 70, No. 3, 283-293

2.      R.A. Cottis, M.A.A. Al-Awadhi, H. Al-Mazeedi, S. Turgoose, Electrochim. Acta 46 (2001): p. 3665.

3.      F. Mansfeld, Z. Sun, Corrosion, 55 (1999) 915.

4.      S.A. Reid, D.A. Eden, US9264824B1, Reino Unido, 2001.

5.      R.A. Cottis et al. / Electrochimica Acta 46 (2001) 3665–3674

Annexx in the Discussion of the paper

      The use of LI to determining corrosion types has been a subject of many discussions among investigators on the data treatment and interpretations using LI [29, 30, 38, 57-59]. Since the mean of the noise data (detrended) would be negligible, the standard deviation and root mean square current noise would converge to the same value and, hence, the LI evaluated from the data would be unity, irrespective of the corrosion type. Cottis [60] indicate that LI for identification of localization of corrosion is unduly influenced by the mean current and hence less reliable. In the present study, and as a first attempt, the LI parameter was estimated. Of course, it is recognized that in the study of stainless steels such as those in the present work, investigation of procedures based on the frequency domain and time-frequency domain deserves further attention.

29.     Suresh G.; U. Kamachi M.S. Electrochemical Noise Analysis of Pitting Corrosion of Type 304L Stainless Steel. Corrosion. 2014, 70 (3), 283-293

30.     Homborg, A.M.; Cottis, R.A.; Mol, J.M.C. An integrated approach in the time, frequency and time-frequency domain for the identification of corrosion using electrochemical noise. Electrochim. Acta. 2016, 222, 627–640.

38.     Mansfeld, F.; Sun, Z. Localization index obtained from electrochemical noise analysis. Corrosion. 1999, 55(10), 915-918

57.     Cottis, R.A.; Al-Awadhi, M.A.A.; Al-Mazeedi, H.; Turgoose S., Measures for the detection of localized corrosion with electrochemical noise. Electrochim. Acta. 2001, 46, 3665-3674

58.     Jurak, T.; Jamali, S.S.; Yue Zhao, Y. Theoretical analysis of electrochemical noise measurement with single substrate electrode configuration and examination of the effect of reference electrodes. Electrochimica Acta 2019, 3011, 377-389

59.     Cottis, R.A. Interpretation of Electrochemical Noise Data. Corrosion. 2001, 57, 3, 265-285

Reference list is very expanded but mostly by quite old articles. There are only 9/54 which are from last 5 years.

Attended:

References have been removed and some of the most current ones have been integrated

Round 2

Reviewer 1 Report

The authors have modified the manuscript and addressed this reviewers concerns pertinent to the previous submission.

Reviewer 3 Report

Dear Authors,

Check the initials and notation in Authors contributions.

Good work! :)

Author Response

Thanks.